# Modeling post-holiday surge in COVID-19 cases in Pennsylvania counties

**Benny Ren**[ID]*, **Wei-Ting Hwang**

Department of Biostatistics, Epidemiology, and Informatics, University of Pennsylvania, Philadelphia, Pennsylvania, United States of America

* bennyren@pennmedicine.upenn.edu

**Data Availability Statement:** The data can be freely accessed from the New York Times GitHub repository (https://github.com/nytimes/covid-19-data).

## Abstract

COVID-19 arrived in the United States in early 2020, with cases quickly being reported in many states including Pennsylvania. Many statistical models have been proposed to understand the trends of the COVID-19 pandemic and factors associated with increasing cases. While Poisson regression is a natural choice to model case counts, this approach fails to account for correlation due to spatial locations. Being a contagious disease and often spreading through community infections, the number of COVID-19 cases are inevitably spatially correlated as locations neighboring counties with a high COVID-19 case count are more likely to have a high case count. In this analysis, we combine generalized estimating equations (GEEs) for Poisson regression, a popular method for analyzing correlated data, with a semivariogram to model daily COVID-19 case counts in 67 Pennsylvania counties between March 20, 2020 to January 23, 2021 in order to study infection dynamics during the beginning of the pandemic. We use a semivariogram that describes the spatial correlation as a function of the distance between two counties as the working correlation. We further incorporate a zero-inflated model in our spatial GEE to accommodate excess zeros in reported cases due to logistical challenges associated with disease monitoring. By modeling time-varying holiday covariates, we estimated the effect of holiday timing on case count. Our analysis showed that the incidence rate ratio was significantly greater than one, 6-8 days after a holiday suggesting a surge in COVID-19 cases approximately one week after a holiday.

## Introduction

COVID-19, a highly contagious respiratory disease, first appeared in China at the end of 2019 and quickly spread across the world [1]. Evidence suggests mask-wearing, and social distancing are effective strategies in containing COVID-19 [2, 3]. During the beginning of the pandemic, local and state governments quickly moved to implement mask mandates, travel restrictions and community containment measures (e.g., shelter in place) to mitigate the spread of the disease [4–6]. However, many Americans still choose to travel and congregate during the pandemic which is heightened during a federal holiday. Due to lack of adherence to public health guidance during the holidays, one should expect to see a surge in COVID-19

**Funding:** WH is supported by National Institute of Environmental Health Sciences grant: P30-ES013508. The funders had no role in study design, data collection and analysis, decision to publish, or preparation of the manuscript. There was no additional external funding received for this study.

**Competing interests:** The authors have declared that no competing interests exist.

cases after a holiday. While many reports reaffirm this hypothesis, they are based on anecdotal evidence such as summary statistics of case counts from moving time windows. There are only a handful epidemiological studies that estimate the association between holiday timing and the number of reported COVID-19 cases [7, 8]. For the first year of the pandemic, we hypothesize that we should see a surge in COVID-19 cases within two weeks after a holiday given that the incubation period for COVID-19 extends up to 14 days, with a median time of 4-5 days from exposure to symptoms onset and adding up to an additional 3 days, either by the PCR test or the instant rapid antigen test, for a positive test to be reported [1, 9–11]. We consider daily case counts between March 20, 2020 to January 23, 2021 to study early pandemic dynamics prior to widespread vaccine distribution, COVID-19 variants and at-home testing. In addition, rigorous disease surveillance and reporting procedures were in place during the beginning of the pandemic resulting in comprehensive infection data.

Poisson count regression with the population size as an offset is a popular approach to model count data and incidence rate. Based on the reporting guidelines of the COVID-19 datasets, there are certain dates such as holidays and weekends that could impact whether cases are being reported [12]. These reporting practices have resulted in excess or structural zeros in the case count, also known as zero-inflation [13]. We also need to consider spatial correlation among county-level COVID-19 case counts because vector-borne and transmissible diseases such as COVID-19, exhibit non-negligible spatial correlation as the movement of people can spread the virus to nearby counties [14]. Furthermore, processes that are confounded by spatially correlated variables are also suited for spatial models; a well-studied example is disease and pollution [15–17]. Thus, we expect to see similar case numbers or trends among neighboring counties [18].

Mixed models are powerful tools for spatial modeling due to its ability to handle a complex spatial correlation structure usually represented as a semivariogram or kriging process [19, 20]. Mixed modeling problems have been addressed from a convex optimization and Bayesian computation perspective [14, 21]. Correlation in spatial epidemiology can also be captured using conditional autoregressive models [22–24]. Inferential summaries from these models assume a correctly specified spatial correlation, otherwise a post-estimation robust covariance can be derived to address misspecified correlation. One such class of robust estimators are the heteroskedasticity-consistent or sandwich estimators which can easily be derived from marginal models and generalized estimating equations (GEEs). Marginal models are flexible alternatives to mixed models when population-level effects are of interest [25, 26]. In addition, formulation of GEEs through the quasi-likelihood provide a convenient set of score equations with well established optimization procedures. As a result, a GEE formulation can be derived to incorporate spatial-temporal relationships, as well as zero-inflation. For its generalizability and robust inference under misspecified spatial correlation, GEEs are an appealing but under utilized tool in spatial epidemiology.

We propose the use of a mixture of marginal models for zero-inflated over-dispersed Poisson regression to model the daily number of new cases in 67 Pennsylvania counties [27]. We combine the zero-inflated Poisson regression with the framework of a transition model to estimate present case counts as a function of previously reported cases [24, 25]. We account for spatial correlation by treating seminvariograms as the working correlation under a generalized estimation equation framework and designate each date as a 'cluster' under the traditional longitudinal nomenclature [26, 28–30]. We propose an Expectation-Solution (ES) algorithm to fit the mixture of marginal models, which conveniently reduces to simpler problems of weighted GEEs and weighted semivariograms [31–35]. We include a combination of past case counts and time-lagged covariates, as well as county-specific and date-specific covariates to model daily rates of new COVID-19 cases following the parameterization of semivariogram working

correlations discussed in [30]. While there are other methods to model spatial temporal data, we elect to use a GEE due to consistent inference under misspecified working correlation, simplicity of formulation and estimation under an imbalance clusters study design [14, 36, 37]. Imbalance clusters occurs in our data because counties report their first case at different dates, staggered entry into the study, when populous counties report cases before rural counties.

The rest of this manuscript is organized as follows. In the Methods section, we outline the zero-inflated Poisson model and the proposed semivariogram model; we detail the zero-inflated GEE for the Poisson counts with excess zeros and describe the estimating procedures for the semivariogram through the ES algorithm and outline the robust sandwich standard error. In the Results section, we analyze the daily COVID-19 cases from 67 Pennsylvania counties.

## Methods

### Zero-inflated Poisson model

Let $Y_{i,t}$ denote the count of COVID-19 cases at county $i$ and date $t$. Often times, a case count of zero is due to logistical issues in reporting, which is known as an excess zero and is unrelated to the Poisson model. A true zero is unrealistic in many situations. For excess zeros, we define the zero-inflation process with Poisson count data as

$$Y_{i,t} \mid Y_{i,t-1} \sim \begin{cases} 0 & \text{with probability } p_{i,t} \\ \text{Poisson}(\mu_{i,t} \mid Y_{i,t-1}) & \text{with probability } 1 - p_{i,t} \end{cases}.$$

That is, if $Y_{i,t}$ is a zero Poisson random variable, it belongs to a Poisson distribution with probability $1-p_{i,t}$, otherwise it is an excess zero with probability $p_{i,t}$ [13]. We denote the latent membership, excess zero indicator, for county $i$ at date $t$ as $Z_{i,t}|Z_{i,t-1} \sim \text{Bernoulli}(p_{i,t}| Z_{i,t-1})$, following a transition model as a function of previous outcomes that is unobservable and treated as missing data.

To model case counts as a rate, we define the Poisson regression model as

$$\mathbb{E}\left[Y_{i,t} \mid Y_{i,t-1} = y_{i,t-1}\right] = \mu_{i,t} = \exp\left(\mathbf{x}_{i,t}^\top \boldsymbol{\beta} + \beta^* \times \log\left(y_{i,t-1}/m_i\right) + \log\left(m_i\right)\right) \quad (1)$$

and (1) can be rewritten as

$$\mathbb{E}\left[\frac{Y_{i,t}}{m_i} \mid Y_{i,t-1} = y_{i,t-1}\right] = \exp\left(\mathbf{x}_{i,t}^\top \boldsymbol{\beta}\right) \left(\frac{y_{i,t-1}}{m_i}\right)^{\beta^*}$$

where $m_i$ is the county population offset term, representing the population at county $i$ but fixed for all dates $t$ and we denote $\mu_{i,t}|Y_{i,t-1}$ as $\mu_{i,t}$ and $p_{i,t}|Z_{i,t-1}$ as $p_{i,t}$ for brevity. We include a transition term of a lagged case incidence $Y_{i,t-1}$ to account for the temporal correlation of counts in the same county. Note that for $y_{i,t-1} = 0$ values must be corrected such that $\log(y_{i,t-1})$ are defined [24]. Our transition term addresses temporal trends and correlation allowing us to treat the residuals from different dates as independent in the working correlation of the marginal model. Here, $\mathbf{x}_{i,t}$ are county and date-specific covariates detailed in our real data analysis.

We can also construct a logistic regression to model the latent membership $Z_{i,t}$ as

$$\mathbb{E}\left[Z_{i,t} \mid Z_{i,t-1} = z_{i,t-1}\right] = p_{i,t} = \text{expit}\left(\mathbf{u}_{i,t}^\top \boldsymbol{\lambda} + \lambda^* \times z_{i,t-1}\right). \quad (2)$$

We also incorporate a transition term, $Z_{i,t-1}$, to account for temporal trends and $\mathbf{u}_{i,t}$ are county and date-specific covariates specifically related to the zero-inflation process.

## Theoretical semivariogram model

Standard zero-inflated Poisson models do not account for the longitudinal nature of the data which is expressed as spatial correlation among counties. Though mixed GLMs can be used to account for spatial correlation as Gaussian random effects, these frameworks involve difficult computation such as the Laplace approximation involving the covariance matrix. Spatial random effects are often modeled using semivariograms which can be viewed as a kriging model in the residual space [14]. Alternatively, we propose a GEE procedure which iteratively updates a working correlation using residuals allowing for a more straight forward estimation procedure.

Semivariograms model the correlation between two locations $i$ and $j$ as a decreasing function of distance. This phenomenon is commonly referred to as Tobler's first law of geography which states: "everything is related to everything else, but near things are more related than distant things." In our study, distance is measured by meters and locations are latitude and longitude coordinates of county centroids. In our GEE procedure, residuals are expected to be correlated based on distance and are used to calculate the semivariogram working correlation. At a given date, county residuals $r_{i,t}$ and $r_{j,t}$, separated by distance $d_{i,j}$, are assumed to follow a theoretical semivariogram $\gamma(d_{i,j}, \tau)$

$$
\begin{aligned}
2\gamma\left(d_{i,j}, \tau\right) &= \mathrm{E}\left[\left(r_{i,t} - r_{j,t}\right)^2\right] \\
&= \mathrm{Var}\left(r_{i,t}\right) + \mathrm{Var}\left(r_{j,t}\right) - 2\mathrm{Cov}\left(r_{i,t}, r_{j,t}\right) \\
&= \phi\left(1 + 1 - 2\mathrm{Corr}\left(r_{i,t}, r_{j,t}\right)\right) \\
\gamma\left(d_{i,j}, \tau\right) &= \phi\left(1 - \exp\left(\frac{-d_{i,j}}{\tau}\right)\right)
\end{aligned}
$$

where $\phi$ is an over-dispersion parameter and $\tau$ is a scale parameter and correlation does not depend on time. Thus, the working correlation matrix between counties $i$ and $j$ at date $t$ is given as

$$
\mathbf{R}_t\left(\tau\right) = \left[\exp\left(\frac{-d_{i,j}}{\tau}\right)\right]_{ij}
$$

and is solely a function of distance $d_{i,j}$ with $\tau$ determining the rate correlation decays as distance increases. We make an isotropic covariance assumption, meaning the semivariogram is a function of the locations only through the Euclidean distance between them.

While standard GEEs treat longitudinal data from individuals as independent clusters, our approach differs by treating dates as independent clusters, counties as our repeated measurements, spatial correlation as the working correlation as summarized in Table 1. After incorporating date-specific covariates and case counts from the previous day, we assume residuals from different dates to be uncorrelated in the working correlation. Fig 1 shows low autocorrelation of residuals, $\mathrm{Corr}(r_{i,t}, r_{i,t-k})$, within each county from a naive standard Poisson generalized linear model described by Eq (1) using a lagged autoregressive predictor ($y_{i,t-1}$) and study

**Table 1. Differences between standard GEE found in longitudinal studies and the proposed GEE.**

|  | Standard GEE | Proposed GEE |
|---|---|---|
| Clusters | Individuals/Patients | Time points (Dates) |
| Repeated Measures | Repeated at different time points | Repeated at different locations |
| Working Correlation | Independent, Exchangeable, AR, etc. | Semivariogram model |

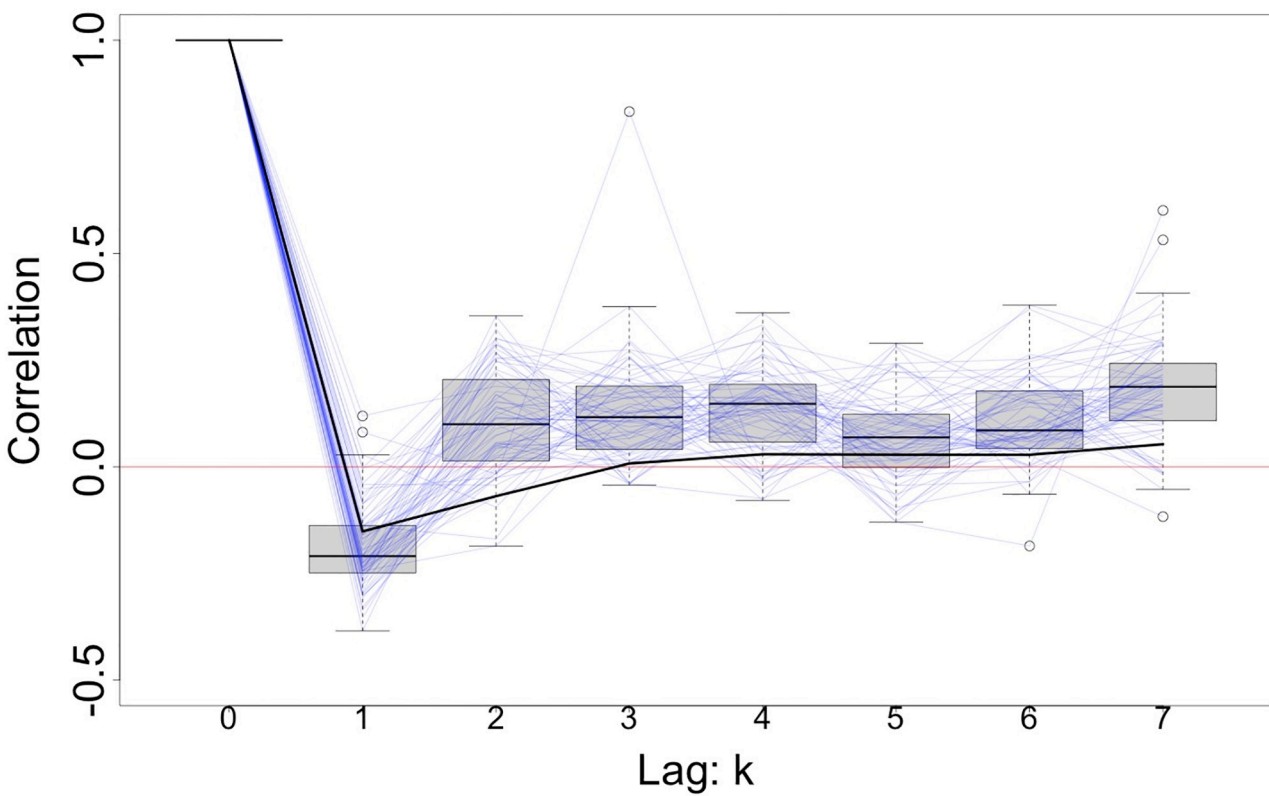

**Fig 1. Box plot of the residual autocorrelations, Corr($r_{i,t}$, $r_{i,t-k}$), for lag $k$ ($k = 1, \ldots 7$) from 67 Pennsylvania counties.** The residuals $r_{i,t}$ were obtained from a Poisson generalized linear model with study covariates and a lagged autoregressive predictor ($y_{i,t-1}$) as outlined in the COVID-19 data analysis section. Light blue lines connect the autocorrelation values of each county. Solid black line connects the autocorrelation values Corr($r_t$, $r_{t-k}$) calculated with residuals from all counties combined.

covariates while assuming no zero-inflation. Our implementation of zero-inflated modeling assumes some observations belong to a structural zero model which further decreases autocorrelation in residuals, suggesting a single lag autoregressive predictor adequately addresses autocorrelation in the residuals. Furthermore, we expect the structural zeros to be related to holiday and weekend timings based on dataset guidelines and case counts to be cumulatively reported after a holiday. As a result, we anticipate holiday and weekend timings to be the major driver of temporal case trajectories in our study. Next, we detail the estimation approach using GEEs and estimation of scale parameter $\tau$ based on procedures from the spatial statistics literature. In many practical settings there is an abundance of time-series data, but limited number of locations; our approach allows the asymptotic results to be driven by the number of dates in the analysis.

## Estimation

**GEE for over-dispersed Poisson counts.** The GEE for the Poisson model is given as

$$Q\left(\boldsymbol{\beta}\right) = \sum_{t=1}^{T} Q_t\left(\boldsymbol{\beta}\right) = \sum_{t=1}^{T} \mathbf{D}_t^{\top}\left(\boldsymbol{\mu}_t\right)\mathbf{V}_t^{-1}\left(\tau_P, \phi\right)\mathbf{W}_t\left(\mathbf{y}_t - \boldsymbol{\mu}_t\right) \tag{3}$$

where

$$\begin{aligned}
\mathbf{V}_t\left(\tau_P, \phi\right) &= \phi \mathbf{A}_t^{1/2} \mathbf{R}_t\left(\tau_P\right) \mathbf{A}_t^{1/2} \\
\mathbf{D}_t\left(\boldsymbol{\mu}_t\right) &= \frac{\partial}{\partial \lambda} \boldsymbol{\mu}_t = \left[\mathbf{x}_{i,t}^\top \mu_{i,t}\right]_i = \mathbf{A}_t \mathbf{X}_t \\
\mathbf{A}_t &= \operatorname{diag}\left(\mu_{1,t}, \mu_{2,t}, \mu_{3,t}, \ldots\right) \\
\mu_{i,t} &= \exp\left(\mathbf{x}_{i,t}^\top \boldsymbol{\beta} + \beta^* \times \log\left(y_{i,t-1}^*/m_i\right) + \log\left(m_i\right)\right)
\end{aligned}$$

with $\mathbf{X}_t = [\mathbf{x}_{1,t}, \mathbf{x}_{2,t}, \mathbf{x}_{3,t}, \ldots]^\top$ and $\boldsymbol{\beta}$ are the design matrix of covariates and vector of regression coefficients related to case counts. Here $Y_{i,t-1}^*$ is the last count greater than zero for county $i$ and $\mathbf{y}_t = [y_{1,t}, y_{2,t}, y_{3,t}, \ldots]^\top$ is a vector representing new COVID-19 cases with corresponding expected values $\boldsymbol{\mu}_t = [\mu_{1,t}, \mu_{2,t}, \mu_{3,t}, \ldots]^\top$. Scale parameter $\tau_P$ is specific to the Poisson model. We incorporated Poisson model membership probabilities, $w_{i,t} = 1-p_{i,t}$, through matrix $\mathbf{W}_t = \operatorname{diag}(w_{1,t}, w_{2,t}, w_{3,t}, \ldots)$, into the weighted GEE. When $w_{i,t} \approx 0$, then the data point is effectively removed from the estimation of Poisson regression parameters. The estimation of these weights is outlined in the Expectation-Solution algorithm in next section. We modify (1) to carry forward the, highly correlated, last non-zero count if $Y_{i,t-1} = 0$. Typically, when $Y_{i,t-1} = 0$ an additional parameter may be introduced in order to ensure $\log\left(y_{i,t-1}^*/m_i\right)$ are defined [24].

In accordance with GEEs, we use residuals to estimate the working correlation but our procedure involves using a parameterization described in the theoretical semivariogram. Over-dispersion is common in COVID-19 case counts due to high variability in reporting practices across counties but is conveniently accounted for with an additional parameter in the GEE. We calculate the dispersion parameter $\phi$ and standard residuals $r_{i,t}$ as

$$\phi = \sum_{t=1}^{T} \sum_{i \in S(t)} \frac{w_{i,t}\left(\frac{y_{i,t}-\mu_{i,t}}{\sqrt{\mu_{i,t}}}\right)^2}{\left(\sum_{t=1}^{T} \sum_{i \in S(t)} w_{i,t}\right) - \operatorname{rank}(\mathbf{X})}$$

$$r_{i,t} = \frac{y_{i,t}-\mu_{i,t}}{\sqrt{\phi \mu_{i,t}}}.$$

where rank$(\mathbf{X})$ is the number of linearly independent covariates and $S(t)$ is the set of observed counties at time $t$. Through the quasi-likelihood, GEEs incorporate a dispersion parameter, resulting in an over-dispersed Poisson regression.

**GEE for excess zeros.** To accommodate for latent membership $Z_{i,t}$, we construct another GEE for estimating $p_{i,t}$, the probability of an excess zero, using the logit link function. This GEE formulation is given as

$$Q(\boldsymbol{\lambda}) = \sum_{t=1}^{T} Q_t(\boldsymbol{\lambda}) = \sum_{t=1}^{T} \mathbf{D}_t^\top\left(\mathbf{p}_t\right) \mathbf{V}_t^{-1}\left(\tau_Z\right)\left(\mathbf{z}_t - \mathbf{p}_t\right) \tag{4}$$

where

$$\begin{aligned}
\mathbf{V}_t\left(\tau_Z\right) &= \mathbf{A}_t^{1/2} \mathbf{R}_t\left(\tau_Z\right) \mathbf{A}_t^{1/2} \\
\mathbf{D}_t\left(\mathbf{p}_t\right) &= \frac{\partial}{\partial \lambda} \mathbf{p}_t = \left[\mathbf{u}_{i,t}^\top p_{i,t}\left(1-p_{i,t}\right)\right]_i = \mathbf{A}_t \mathbf{U}_t \\
\mathbf{A}_t &= \operatorname{diag}\left(p_{1,t}(1-p_{1,t}), p_{2,t}\left(1-p_{2,t}\right), \ldots\right) \\
p_{i,t} &= \operatorname{expit}\left(\mathbf{u}_{i,t}^\top \boldsymbol{\lambda} + \lambda^* \times \mathbb{I}\left(y_{i,t-1} = 0\right)\right)
\end{aligned}$$

and $\tau_Z$ is scale parameter for the excess zero model. Here $\mathbf{U}_t$ and $\boldsymbol{\lambda}$ are the covariates and

regression coefficients for modeling the excess zero. Because we do not have the complete data, in order to account for serial correlation among excess zeros, we replace the transition term $Z_{i,t-1}$ in (2) with an indicator of zero cases on the previous day, $\mathbb{I}\left(y_{i,t-1}=0\right)$ in order capture temporal trends. In practice, $Z$ is often an imbalanced binary outcome where over parameterization of covariates can cause numerical instability when fitting the model. Therefore, it has been suggested that it be fitted with a parsimonious set of covariates [27].

We calculate standard Pearson residuals as

$$e_{i,t} = \frac{z_{i,t} - p_{i,t}}{\sqrt{p_{i,t}\left(1 - p_{i,t}\right)}}$$

which are used for estimating the working correlation using a semivariogram as described in the next section.

**Estimation of $\tau$ through the empirical semivariogram.** After each Newton Raphson update of the regression coefficients $\boldsymbol{\beta}$, $\boldsymbol{\lambda}$. We also update the working correlation by estimating the scale parameter $\tau$, using the empirical semivariogram. In this section we outline procedures for modeling semivariograms and their computational implementation. First, we calculate the empirical semivariogram by grouping pairs of residuals into $U$ bins, based on equally spaced intervals of distance. Each bin of residuals will be used to calculate an empirical estimate for the midpoint of the bin denoted as $d_1, d_2, d_3, \ldots, d_U$ with the corresponding intervals denoted as $d_u \pm \delta$. After we obtain the empirical semivariogram $\hat{\gamma}\left(d_u\right)$, we estimate $\tau$ from the theoretical semivariogram model through a weighted least squares approach that minimizes the squared difference between the theoretical and empirical semivariograms. Using the procedure detailed in [38], the weighted least squares solution is given as

$$\tau = \underset{\tau}{\operatorname{argmin}} \sum_{d_u \in d_1, d_2, \ldots, d_U} \frac{\sum_{t=1}^{T} \sum_{d_{ij} \in d_u \pm \delta} w_{i,t} w_{j,t}}{\left[\gamma\left(d_u, \tau\right)\right]^2} \left[\hat{\gamma}\left(d_u\right) - \gamma\left(d_u, \tau\right)\right]^2 \tag{5}$$

where weights $w_{i,t}$ and $w_{j,t}$ replace the sample size in our case. The weighted least squares approach is computationally fast and yields robust estimates. Placing the theoretical semivariogram in the denominator down weights the influence of observed correlations separated by large distances.

We denote the solution of (5), as a function of the residuals $r_{i,t}$, $e_{i,t}$ and weights by $\tau_P = G(\mathbf{r}, \mathbf{W})$ and $\tau_Z = G(\mathbf{e}, \mathbf{I})$ for the Poisson and excess zero model, respectively. Recall, we use separate GEEs for the Poisson and excess zeros, to account for the spatial correlation through a mixture of marginal (zero-inflated) model. We follow some common practices regarding semivariogram models; we calculate the empirical semivariogram using pairwise distances that are less than half the maximum distance, we also bound $\tau \in (0, \max_{i,j}(d_{i,j})/3)$ so that $\exp(-3) \approx 0.05$, i.e., the correlation associated with maximum distance, is upper bounded at 0.05. In our case, $\max_{i,j}(d_{i,j}) = 592318.3$ meters. Next, we outline options for computing the empirical semivariogram $\hat{\gamma}\left(d_u\right)$.

**Empirical semivariograms.** The standard moment estimate of the empirical semivariogram is given as

$$\hat{\gamma}\left(d_u\right) = \frac{\sum_{t=1}^{T} \sum_{d_{ij} \in d_u \pm \delta} w_{i,t} w_{j,t} \left(r_{i,t} - r_{j,t}\right)^2}{2 \sum_{t=1}^{T} \sum_{d_{ij} \in d_u \pm \delta} w_{i,t} w_{j,t}}.$$

[39] presented an alternative estimate that is robust

$$
\hat{\gamma}_{CH}\left(d_u\right) = \frac{\left(\dfrac{\sum_{t=1}^{T} \sum_{d_{ij} \in d_u \pm \delta} w_{i,t} w_{j,t} \left|r_{i,t} - r_{j,t}\right|^{1/2}}{\sum_{t=1}^{T} \sum_{d_{ij} \in d_u \pm \delta} w_{i,t} w_{j,t}}\right)^{4}}{0.914 + \dfrac{0.988}{\sum_{t=1}^{T} \sum_{d_{ij} \in d_u \pm \delta} w_{i,t} w_{j,t}}}.
$$

[40, 41] presented median based approaches with 50% breakpoint [39–41]. Being that estimating $\tau$ is a sub-routine in the ES algorithm, we elect to use the approach from [39] as our empirical semivariogram: $\hat{\gamma}_{CH}\left(d_u\right)$ for robustness considerations. We denote the corresponding $\tau$ estimates as $G_{CH}(\mathbf{r}, \mathbf{W})$, and $G_{CH}(\mathbf{e}, \mathbf{I})$ in our analysis. The empirical semivariogram is calculated in parallel and $\tau$ is estimated numerically using weighted least squares and the `optim` package in R [42].

## Expectation-Solution algorithm

Following [27] we construct an Expectation-Solution algorithm for our zero-inflated regression models as a mixture of marginal models [27, 31]. The ES algorithm is a modification to the well-known Expectation-Maximization (EM) algorithm, where the M-step is replaced by the solution from the GEE [43]. We further modify the working correlation to follow a spatial correlation structure which is estimated using a semivariogram [30].

**E-step.** In the E-step, we update the unobserved $\mathbf{Z}$. The conditional expectation is given as:

$$
z_{i,t}^{(s+1)} = \left[\frac{p_{i,t}^{(s)}}{p_{i,t}^{(s)} + \left(1 - p_{i,t}^{(s)}\right) f\left(y_{i,t} \mid \mu_{i,t}^{(s)}\right)}\right] \mathbb{I}\left(y_{i,t} = 0\right) \tag{6}
$$

$$
w_{i,t}^{(s+1)} = 1 - z_{i,t}^{(s+1)} \tag{7}
$$

where $w_{i,t}^{(s+1)}$ are updated Poisson model membership weights. We denote the iteration number of the ES algorithm as $s$.

**S-step.** The S-step replaces the maximization step in the EM algorithm. In the S-step, we iteratively estimate Poisson model parameters $\boldsymbol{\beta}$, $\tau_P$ and the excess zero model parameters $\lambda$, $\tau_Z$. The iterative updates for $\boldsymbol{\beta}$ using Newton Raphson are given as:

$$
\boldsymbol{\beta}^{(k+1)} = \boldsymbol{\beta}^{(k)} + \mathbf{H}^{-1}\left(\boldsymbol{\mu}^{(k)}, \tau_P^{(k)}, \phi^{(k)}\right) \mathbf{Q}\left(\boldsymbol{\mu}^{(k)}, \tau_P^{(k)}, \phi^{(k)}, \mathbf{W}^{(s+1)}\right) \tag{8}
$$

$$
\mathbf{H}^{-1}\left(\boldsymbol{\mu}^{(k)}, \tau_P^{(k)}, \phi^{(k)}\right) = \left(\sum_{t=1}^{T} \left[\mathbf{D}_t\left(\boldsymbol{\mu}_t^{(k)}\right)\right]^{\top} \left[\mathbf{V}_t\left(\tau_P^{(k)}, \phi^{(k)}\right)\right]^{-1} \left[\mathbf{D}_t\left(\boldsymbol{\mu}_t^{(k)}\right)\right]\right)^{-1}
$$

$$
\mathbf{Q}\left(\boldsymbol{\mu}^{(k)}, \tau_P^{(k)}, \phi^{(k)}, \mathbf{W}^{(s+1)}\right) = \sum_{t=1}^{T} \left[\mathbf{D}_t\left(\boldsymbol{\mu}_t^{(k)}\right)\right]^{\top} \left[\mathbf{V}_t\left(\tau_P^{(k)}, \phi^{(k)}\right)\right]^{-1} \mathbf{W}_t^{(s+1)}\left(\mathbf{y}_t - \boldsymbol{\mu}_t^{(k)}\right)
$$

and $\tau_P^{(k+1)}$ is updated with (5) using $\hat{\gamma}_{CH}\left(d_u\right)$. We denote the update as

$$
\tau_P^{(k+1)} = G_{CH}\left(\mathbf{r}^{(k+1)}, \mathbf{W}^{(s+1)}\right) \tag{9}
$$

where

$$\phi^{(k+1)} = \sum_{t=1}^{T} \sum_{i \in i \in S\ (t)} \frac{w_{i,t}^{(s+1)} \left( \frac{y_{i,t} - \mu_{i,t}^{(k+1)}}{\sqrt{\mu_{i,t}^{(k+1)}}} \right)^2}{\left( \sum_{t=1}^{T} \sum_{i \in S\ (t)} w_{i,t}^{(s+1)} \right) - \text{rank}\,(\mathbf{X})}$$

$$r_{i,t}^{(k+1)} = \frac{y_{i,t} - \mu_{i,t}^{(k+1)}}{\sqrt{\phi^{(k+1)} \mu_{i,t}^{(k+1)}}}.$$

The iteration number of the S-step are denoted with $k$. We repeat (8) and (9) until convergence to obtain $\boldsymbol{\beta}^{(s+1)}$, a new iteration in the greater ES algorithm.

We apply the same procedure in the S-step for excess zero model parameters: $\lambda$ and $\tau_Z$

$$\lambda^{(k+1)} = \lambda^{(k)} + \mathbf{H}^{-1}\left( \mathbf{p}^{(k)}, \tau_Z^{(k)} \right) \mathbf{Q}\left( \mathbf{p}^{(k)}, \tau_Z^{(k)}, \mathbf{z}^{(s+1)} \right) \tag{10}$$

$$\mathbf{H}^{-1}\left( \mathbf{p}^{(k)}, \tau_Z^{(k)} \right) = \left( \sum_{t=1}^{T} \left[ \mathbf{D}_t\left( \mathbf{p}_t^{(k)} \right) \right]^{\top} \left[ \mathbf{V}_t\left( \tau_Z^{(k)} \right) \right]^{-1} \left[ \mathbf{D}_t\left( \mathbf{p}_t^{(k)} \right) \right] \right)^{-1}$$

$$\mathbf{Q}\left( \mathbf{p}^{(k)}, \tau_Z^{(k)}, \mathbf{z}^{(s+1)} \right) = \sum_{t=1}^{T} \left[ \mathbf{D}_t\left( \mathbf{p}_t^{(k)} \right) \right]^{\top} \left[ \mathbf{V}_t\left( \tau_Z^{(k)} \right) \right]^{-1} \left( \mathbf{z}_t^{(s+1)} - \mathbf{p}_t^{(k)} \right)$$

$$\tau_Z^{(k+1)} = G_{CH}\left( \mathbf{e}^{(k+1)}, \mathbf{W} = \mathbf{I} \right) \tag{11}$$

where

$$e_{i,t}^{(k+1)} = \frac{z_{i,t}^{(s+1)} - p_{i,t}^{(k+1)}}{\sqrt{p_{i,t}^{(k+1)}\left( 1 - p_{i,t}^{(k+1)} \right)}}$$

and the weight matrices are replaced by the identity matrix. Eqs (10) and (11) are repeated until convergence to obtain $\lambda^{(s+1)}$. For the ES algorithm, we iteratively update the E-step: $\mathbf{z}^{(s)}$, $\mathbf{W}^{(s)}$, and the S-step: $\boldsymbol{\beta}^{(s)}$, $\lambda^{(s)}$ until convergence.

## Inference on coefficients

Using the heteroskedasticity-consistent or robust sandwich estimator for standard errors,

$$\text{cov}\,(\boldsymbol{\beta}) = \left( \sum_{t=1}^{T} \mathbf{D}_t^{\top} \mathbf{V}_t^{-1} \mathbf{W}_t \mathbf{D}_t \right)^{-1} \left( \sum_{t=1}^{T} Q_t\,(\boldsymbol{\beta}) Q_t^{\top}\,(\boldsymbol{\beta}) \right) \left( \sum_{t=1}^{T} \mathbf{D}_t^{\top} \mathbf{V}_t^{-1} \mathbf{W}_t \mathbf{D}_t \right)^{-1}$$

we derive confidence intervals for $\boldsymbol{\beta}$; we have consistent estimates even under misspecified working correlations which are spatial correlations in our case [31]. We proposed a reasonable spatial correlation model but retain reliable inference even in situations when the correlation model is incorrect. Analogously, we can also calculate the covariance for $\lambda$ with the equivalent formulation without the weight matrices, $\mathbf{W}_t$, using the score Eq (4). We use the asymptotic distribution, $\hat{\boldsymbol{\beta}} \sim \text{N}(\boldsymbol{\beta}, \text{cov}(\boldsymbol{\beta}))$ for inference, substituting the final estimates from the ES algorithm.

## Results

### Data

**Poisson model covariates.** Daily reported cases for 67 counties in Pennsylvania from March 20, 2020 to January 23, 2021 were obtained from the New York Times GitHub repository, https://github.com/nytimes/covid-19-data [12]. We restrict our study dates to be prior to widespread vaccine distribution, COVID-19 variants and at-home testing. The New York Times dataset includes confirmed cases from PCR tests as well as probable cases. Probable cases are derived from a set of testing, symptoms and exposure criteria recommended by the Council of State and Territorial Epidemiologists and was also adopted by the Center of Disease Control on April 14, 2020 [44]. Data collection for a county starts after the first recorded case and we exclude the first 14 days of documented cases per county as reporting is often inconsistent during the start of data collection [45]. This also ensures carried forward imputation values: $y^*_{i,t-1} > 0$ and $\log(y^*_{i,t-1}/m_i)$ are all defined.

Our primary goal is to understand the relationship between the timing of a holiday and daily new case counts. We consider federal holidays: New Years, Martin Luther King, George Washington Birthday, Good Friday, Memorial, Independence, Labor, Columbus, Veterans, Thanksgiving, Christmas and election day (November 3rd, 2020) [46]. Federal holidays are dates which a large proportion of the population are not working, enabling congregation and subsequent transmission of COVID-19. We create a vector of holiday binary {1, 0} indicators for whether a day in the last 2 weeks was a holiday in the Poisson regression model from today to the prior 14 days. For example, the count recorded on January 1st, the associated 15 holiday lag indicators are (1, 0, 0, 0, 0, 0, 0, 1, 0, 0, 0, 0, 0, 0, 0) for dates: New Years, Dec 31, Dec 30, Dec 29, Dec 28, Dec 27, Dec 26, Christmas, . . ., Dec 18. We abbreviate the indicator for a holiday being $k$ days in the past as, Holiday Lag $k$: HL$k$. Another example, for the count data on January 2nd, the indicators are (0, 1, 0, 0, 0, 0, 0, 0, 1, 0, 0, 0, 0, 0, 0). Therefore, the regression coefficients of our holiday indicators can inform whether there are increased cases (i.e., surge) with incidence rate ratios greater than 1, in relation to the presence and timing of a holiday within the prior 2 weeks. If the hypothesis of a post-holiday surge in COVID-19 cases is true, we expect to see several incidence rate ratios (IRRs) associated with HL$k$ to be significantly greater than 1 after a holiday.

However, to account for the association with the day of the week, we include an indicator for the day of the week with Wednesday as the baseline. Our regression analysis also controls for other factors or covariates that may be associated with the incidence of COVID-19 cases as well as excess zeros in reporting. For covariates associated with case incidence, there's growing evidence suggesting that warm and wet climate conditions seem to reduce spread of COVID-19 [47]. Precipitation and maximum temperature data from daily summaries was obtained from National Oceanic and Atmospheric Administration (NOAA) web service were included as additional covariates [48]. Maximum daily temperature and daily precipitation from weather stations across Pennsylvania were downloaded from the NOAA web service. Using latitude and longitude for each county centroid and weather station, county-specific weather data was interpolated using weather data from the same date. Interpolation was carried out using the `VIM` R package, median of $K$ nearest neighbors (*K-nn*), with $K = 5$ [49]. Precipitation was further binarized into a {0, 1} indicator based on whether or not it rained that day. Similar to the holiday covariates, we also construct 15 lag covariate indicators for precipitation in the prior 2 weeks. We also construct 15 continuous covariates of Fahrenheit temperature values for the prior 2 weeks. Following the convention of abbreviating Holiday Lag: HL$k$, we abbreviate the covariates for temperature as TL$k$ and precipitation as PL$k$.

Other covariates included for the Poisson regression consisted of well-known demographic factors that affect health outcomes [50, 51]. Demographic information from the U.S. census and 2016 American Community Survey were also included in the model as county-specific covariates. For each county, quintiles for population density, percentage of the population living in poverty, log median household income, log median house value, percentage of Black residents, percentage of Hispanic residents, percentage of the adult population with less than high school (HS) education, and percentage of owner-occupied housing were included as county-specific covariates. The data was obtained using the same source outlined in [51].

**Zero-inflation covariates.** The presence of excess zeros can bias coefficient estimates in the negative direction if ignored. A non-negligible 21% of the reported case counts are zeros. Therefore, we propose a parsimonious model for $Z$ by considering only a subset of covariates from the main Poisson regression model such that zero-inflation is accounted for based on well-known relationships associated with reporting zero cases. Based on the COVID-19 case reporting guidelines, covariates associated with excess zeros include holidays and weekends. For example, there tends to be a gap in reporting during weekends and holidays, suggesting case count reporting is more likely to occur on business days which does not reflect the reality of the pandemic and back logged cases tend to be lumped into the next business day's case count.

Indicators for tomorrow, today and yesterday being a holiday are included model for $Z$. As an example, for outcome data recorded on December 31st, the covariates will be $(1, 0, 0)$. An indicator for weekend is also included. In addition, certain county-specific covariates, percentage of population in poverty and percentage of owner-occupied housing are also included as covariates. Owner-occupied housing rate is an indicator of urban-rural dichotomy, with rural counties having a high rate of owner-occupied housing. Poverty rate and owner-occupied housing rate, as socioeconomic markers, are potential confounders for zero-inflation since the economy of a county may determine the resources available for municipal services such as case reporting. We also include $\mathbb{I}(y_{i,t-1} = 0)$ an indicator for whether the previous day was a zero as a covariate to account for temporal trends.

## COVID-19 data analysis

In addition to the Poisson model, we fitted a negative binomial hurdle model to capture the severity of a COVID-19 outbreak conditional that cases counts are greater than zero [52, 53]. The hurdle model fits a negative binomial regression truncated to the positive case counts, $\{Y_{i,t} : Y_{i,t} > 0\}$ using the same covariates as (3) and models outcomes $\mathbb{I}(Y_{i,t} = 0)$ with a logistic regression with the same covariates as (4). Robust sandwich estimators for the hurdle model were obtained using the `sandwich` R package [54, 55].

We fitted the following four models: 1.) Poisson generalized linear model without spatial correlation, without over-dispersion and without zero-inflation, we denoted as GLM, 2.) negative binomial hurdle model without spatial correlation, we denoted as Hurdle-NB, 3.) over-dispersed Poisson marginal model, with spatial correlation, but without zero-inflation, we denote this as GEE-OP, 4.) a zero-inflated over-dispersed Poisson marginal model as described by the Expectation-Solution algorithm, which we denote this as GEE-ZIOP.

The GEE-ZIOP results in Fig 2 and Table 2, shows the estimated incidence rate ratio (and 95% confidence interval) of 1.41 (1.07, 1.85); 1.37 (1.09, 1.72); 1.3 (1.08, 1.57); at 6, 7, 8 respective days after a holiday. At 6 to 8 days after a holiday we expect there to be a surge with 1.3 to 1.41 times as many cases as there would be without a holiday. Intervals for IRR estimates were plotted for all models in Fig 2 and showed that IRR estimates associated with the GEE-ZIOP model at the surge was greater than GLM, Hurdle-NB and GEE-OP models. All four models

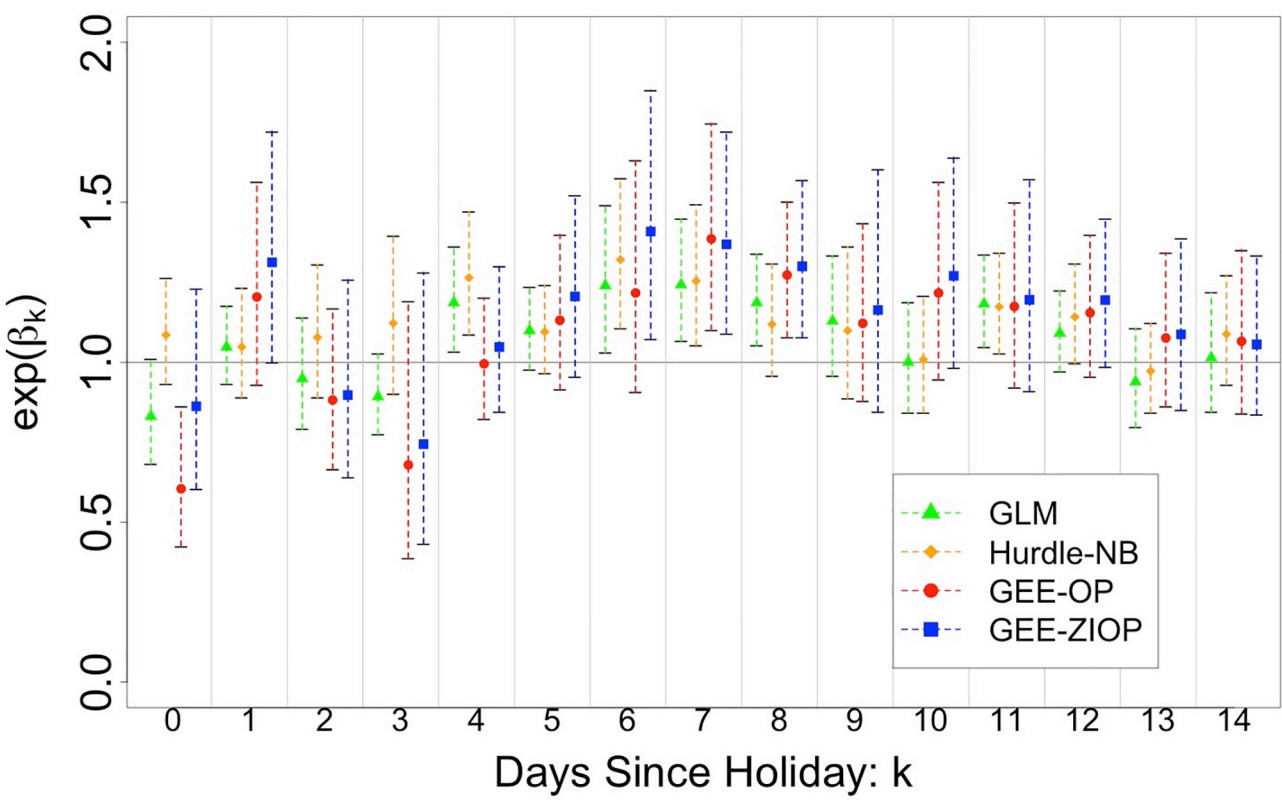

**Fig 2. Incidence rate ratio with robust 95% confidence interval for holiday effects.** Estimated IRRs corresponding to the 15 holiday covariates, for each model. 95% confidence intervals were calculated using robust standard errors.

yield maximum IRR estimates at 6-8 days, suggesting that a surge in case counts occurs about 6-8 days after a holiday.

We did not find a similar time-varying effects of temperature or precipitation (results can be found in Table 3 & 4) and most of the IRRs associated with TL$k$ and PL$k$ covariates were not significant. From the results of the GEE models in Table 5, we observed a significant increase in case incidence on Mondays, which aligns with the dataset guideline. Cases that were not reported during the weekend, possibly due to logistical issues, were reported on Monday instead. Additional county demographic effects can be found in Table 5. Notably, counties with a high proportion African-Americans were associated with increased COVID-19 incidence in all four models, with significant effects in the Hurdle-NB and GEE-ZIOP models. In addition, high median household income counties were associated with decreased COVID-19 incidence, in three of the four model, with significant effect in the GEE-OP and GEE-ZIOP models. Counties with low rate of high school education attainment were also associated with increased COVID-19 incidence in all four models, albeit not always at a significant level. Although many county demographic effects varied across the four models, a number of social and demographic effects from our analysis aligned with results from the literature [56, 57].

Our results for modeling zero counts can be found in Table 6. The zero-inflated model differs from a hurdle model by treating zero counts probabilistically through a mixture model, while the hurdle model treats every zero count as a success in a Bernoulli process. For our data, we believe zeros may originate from the Poisson model or be superficially generated due to logistical issues in the reporting and a mixture model is more appropriate. However, a

**Table 2. Intercept and holiday coefficients with 95% confidence intervals based on robust standard errors.**

|  | GLM | | Hurdle-NB | | GEE-OP | | GEE-ZIOP | |
|---|---|---|---|---|---|---|---|---|
|  | $\beta$ | CI | $\beta$ | CI | $\beta$ | CI | $\beta$ | CI |
| Intercept | -1.9081 | (-3.503, -0.313) | 0.8293 | (-4.861, 6.52) | -0.6658 | (-2.576, 1.244) | -0.2922 | (-2.286, 1.702) |
| HL0 | -0.1852 | (-0.382, 0.012) | 0.0819 | (-0.069, 0.233) | -0.5037 | (-0.857, -0.15) | -0.1482 | (-0.504, 0.208) |
| HL1 | 0.0460 | (-0.069, 0.161) | 0.0465 | (-0.116, 0.209) | 0.1855 | (-0.075, 0.446) | 0.2716 | (0, 0.543) |
| HL2 | -0.0525 | (-0.236, 0.13) | 0.0746 | (-0.118, 0.267) | -0.1261 | (-0.408, 0.156) | -0.1084 | (-0.446, 0.229) |
| HL3 | -0.1139 | (-0.255, 0.027) | 0.1150 | (-0.103, 0.333) | -0.3874 | (-0.95, 0.175) | -0.2959 | (-0.839, 0.247) |
| HL4 | 0.1706 | (0.032, 0.309) | 0.2342 | (0.083, 0.385) | -0.0053 | (-0.194, 0.183) | 0.0464 | (-0.168, 0.261) |
| HL5 | 0.0942 | (-0.023, 0.212) | 0.0904 | (-0.036, 0.216) | 0.1231 | (-0.09, 0.336) | 0.1867 | (-0.046, 0.42) |
| HL6 | 0.2146 | (0.029, 0.4) | 0.2781 | (0.101, 0.455) | 0.1958 | (-0.097, 0.489) | 0.3424 | (0.07, 0.615) |
| HL7 | 0.2167 | (0.064, 0.369) | 0.2262 | (0.052, 0.4) | 0.3260 | (0.094, 0.558) | 0.3135 | (0.085, 0.542) |
| HL8 | 0.1704 | (0.05, 0.291) | 0.1118 | (-0.045, 0.269) | 0.2408 | (0.075, 0.406) | 0.2620 | (0.074, 0.45) |
| HL9 | 0.1213 | (-0.045, 0.287) | 0.0941 | (-0.121, 0.309) | 0.1146 | (-0.131, 0.36) | 0.1511 | (-0.169, 0.472) |
| HL10 | 0.0002 | (-0.172, 0.172) | 0.0080 | (-0.171, 0.187) | 0.1957 | (-0.055, 0.446) | 0.2389 | (-0.017, 0.494) |
| HL11 | 0.1673 | (0.045, 0.289) | 0.1597 | (0.026, 0.293) | 0.1604 | (-0.084, 0.405) | 0.1780 | (-0.096, 0.452) |
| HL12 | 0.0870 | (-0.029, 0.203) | 0.1322 | (-0.003, 0.268) | 0.1434 | (-0.048, 0.334) | 0.1774 | (-0.016, 0.371) |
| HL13 | -0.0634 | (-0.228, 0.101) | -0.0282 | (-0.172, 0.115) | 0.0729 | (-0.148, 0.294) | 0.0834 | (-0.161, 0.328) |
| HL14 | 0.0136 | (-0.169, 0.196) | 0.0841 | (-0.073, 0.241) | 0.0632 | (-0.175, 0.301) | 0.0540 | (-0.18, 0.288) |

GEE-OP: $\tau_P$ = 156552.2, $\phi$ = 30.0776 GEE-ZIOP: $\tau_P$ = 114791.4, $\phi$ = 30.84434

GLM: Poisson generalized linear model, without over-dispersion, without zero-inflation, without spatial correlation

Hurdle-NB: negative binomial Hurdle model, without spatial correlation

GEE-OP: over-dispersed Poisson marginal model, without zero-inflation, with spatial correlation

GEE-ZIOP: zero-inflated over-dispersed Poisson marginal model, with spatial correlation

**Table 3. Temperature coefficients and 95% confidence intervals.**

|  | GLM | | Hurdle-NB | | GEE-OP | | GEE-ZIOP | |
|---|---|---|---|---|---|---|---|---|
|  | $\beta$ | CI | $\beta$ | CI | $\beta$ | CI | $\beta$ | CI |
| TL0 | -0.0018 | (-0.006, 0.002) | -0.0061 | (-0.01, -0.002) | 0.0008 | (-0.005, 0.007) | -0.0002 | (-0.005, 0.005) |
| TL1 | -0.0030 | (-0.008, 0.003) | -0.0010 | (-0.006, 0.004) | -0.0067 | (-0.013, -0.001) | -0.0065 | (-0.012, -0.001) |
| TL2 | -0.0016 | (-0.007, 0.003) | -0.0021 | (-0.007, 0.003) | -0.0029 | (-0.008, 0.003) | -0.0033 | (-0.009, 0.002) |
| TL3 | -0.0085 | (-0.014, -0.003) | -0.0076 | (-0.013, -0.002) | -0.0043 | (-0.011, 0.002) | -0.0032 | (-0.009, 0.003) |
| TL4 | 0.0026 | (-0.001, 0.007) | -0.0016 | (-0.007, 0.004) | 0.0034 | (-0.003, 0.01) | 0.0021 | (-0.004, 0.008) |
| TL5 | -0.0002 | (-0.005, 0.004) | 0.0030 | (-0.003, 0.009) | -0.0042 | (-0.01, 0.002) | -0.0037 | (-0.009, 0.002) |
| TL6 | -0.0018 | (-0.006, 0.003) | -0.0017 | (-0.007, 0.003) | 0.0012 | (-0.004, 0.007) | 0.0008 | (-0.004, 0.006) |
| TL7 | -0.0021 | (-0.006, 0.002) | -0.0029 | (-0.007, 0.001) | -0.0014 | (-0.007, 0.004) | -0.0026 | (-0.007, 0.002) |
| TL8 | 0.0014 | (-0.003, 0.006) | 0.0005 | (-0.005, 0.006) | -0.0025 | (-0.008, 0.003) | -0.0030 | (-0.008, 0.002) |
| TL9 | -0.0038 | (-0.009, 0.001) | -0.0023 | (-0.008, 0.003) | -0.0027 | (-0.008, 0.003) | -0.0016 | (-0.007, 0.004) |
| TL10 | -0.0020 | (-0.006, 0.002) | -0.0038 | (-0.009, 0.001) | -0.0004 | (-0.005, 0.005) | -0.0007 | (-0.006, 0.004) |
| TL11 | 0.0011 | (-0.003, 0.005) | 0.0037 | (-0.001, 0.008) | -0.0028 | (-0.008, 0.002) | -0.0029 | (-0.008, 0.002) |
| TL12 | 0.0019 | (-0.003, 0.007) | 0.0019 | (-0.003, 0.007) | 0.0016 | (-0.004, 0.007) | 0.0019 | (-0.004, 0.007) |
| TL13 | -0.0015 | (-0.006, 0.003) | -0.0060 | (-0.01, -0.002) | 0.0012 | (-0.004, 0.007) | 0.0003 | (-0.005, 0.006) |
| TL14 | 0.0027 | (-0.001, 0.007) | 0.0054 | (0.002, 0.009) | 0.0021 | (-0.003, 0.007) | 0.0031 | (-0.002, 0.008) |

**Table 4. Precipitation coefficients and 95% confidence intervals.**

|  | GLM | | Hurdle-NB | | GEE-OP | | GEE-ZIOP | |
|---|---|---|---|---|---|---|---|---|
|  | β | CI | β | CI | β | CI | β | CI |
| PL0 | -0.0176 | (-0.067, 0.032) | -0.0641 | (-0.116, -0.012) | 0.0482 | (-0.025, 0.121) | 0.0294 | (-0.031, 0.089) |
| PL1 | -0.0158 | (-0.072, 0.04) | 0.0182 | (-0.034, 0.071) | -0.0825 | (-0.171, 0.006) | -0.0775 | (-0.161, 0.006) |
| PL2 | -0.0142 | (-0.07, 0.041) | -0.0185 | (-0.072, 0.035) | 0.0049 | (-0.079, 0.089) | 0.0236 | (-0.052, 0.099) |
| PL3 | 0.0081 | (-0.046, 0.062) | -0.0275 | (-0.081, 0.026) | -0.0116 | (-0.073, 0.05) | -0.0011 | (-0.056, 0.054) |
| PL4 | -0.0246 | (-0.082, 0.033) | -0.0146 | (-0.071, 0.041) | 0.0243 | (-0.045, 0.094) | 0.0079 | (-0.06, 0.076) |
| PL5 | -0.0743 | (-0.128, -0.02) | -0.0968 | (-0.148, -0.045) | -0.0581 | (-0.138, 0.021) | -0.0730 | (-0.152, 0.006) |
| PL6 | 0.0212 | (-0.032, 0.074) | -0.0196 | (-0.079, 0.04) | 0.0238 | (-0.044, 0.092) | -0.0220 | (-0.09, 0.046) |
| PL7 | -0.0885 | (-0.143, -0.034) | -0.0714 | (-0.129, -0.014) | -0.0306 | (-0.106, 0.045) | -0.0459 | (-0.115, 0.023) |
| PL8 | -0.0359 | (-0.093, 0.021) | -0.0231 | (-0.076, 0.03) | -0.0503 | (-0.137, 0.036) | -0.0413 | (-0.119, 0.037) |
| PL9 | 0.0258 | (-0.034, 0.086) | 0.0113 | (-0.045, 0.068) | 0.0263 | (-0.051, 0.104) | 0.0483 | (-0.021, 0.118) |
| PL10 | -0.0635 | (-0.126, -0.001) | -0.0724 | (-0.128, -0.017) | -0.0952 | (-0.175, -0.016) | -0.0790 | (-0.145, -0.012) |
| PL11 | 0.0577 | (-0.005, 0.12) | 0.0177 | (-0.035, 0.071) | 0.0357 | (-0.031, 0.103) | 0.0301 | (-0.036, 0.096) |
| PL12 | -0.0633 | (-0.117, -0.01) | -0.0336 | (-0.084, 0.016) | -0.0090 | (-0.093, 0.075) | -0.0234 | (-0.095, 0.048) |
| PL13 | 0.0535 | (-0.003, 0.11) | -0.0262 | (-0.083, 0.03) | 0.0269 | (-0.047, 0.1) | -0.0266 | (-0.099, 0.046) |
| PL14 | -0.0323 | (-0.09, 0.026) | -0.0800 | (-0.128, -0.032) | -0.0047 | (-0.073, 0.063) | -0.0205 | (-0.08, 0.039) |

hurdle model is equivalent to a zero-inflated model when zero counts from the underlying Poisson model are rare and serve as a useful sensitivity analysis under different modeling assumptions. Counties with a high rate of owner occupied housing were significantly associated with an increased probability of reporting zero cases in both models. Owner occupied housing may reflect the urban-rural dichotomy of counties, with rural counties having a high rate of owner occupied housing and, by extension, conclude that rural counties have an

**Table 5. Day of week and county demographic coefficients and 95% confidence intervals based on robust standard errors.**

|  | GLM | | Hurdle-NB | | GEE-OP | | GEE-ZIOP | |
|---|---|---|---|---|---|---|---|---|
|  | β | CI | β | CI | β | CI | β | CI |
| Thursday | 0.1402 | (0.033, 0.247) | 0.1073 | (-0.009, 0.224) | 0.1787 | (-0.031, 0.389) | 0.1864 | (-0.027, 0.4) |
| Friday | -0.0154 | (-0.133, 0.102) | -0.0382 | (-0.149, 0.073) | 0.1256 | (-0.06, 0.311) | 0.1618 | (-0.032, 0.356) |
| Saturday | -0.0789 | (-0.176, 0.018) | -0.0135 | (-0.116, 0.089) | -0.0976 | (-0.274, 0.079) | 0.1794 | (-0.022, 0.381) |
| Sunday | -0.2833 | (-0.407, -0.16) | -0.1567 | (-0.283, -0.03) | -0.2987 | (-0.498, -0.099) | 0.0710 | (-0.158, 0.3) |
| Monday | -0.0064 | (-0.134, 0.121) | -0.1910 | (-0.309, -0.072) | 0.3323 | (0.13, 0.535) | 0.3746 | (0.163, 0.586) |
| Tuesday | 0.1072 | (0.001, 0.213) | 0.1213 | (0.006, 0.237) | 0.1406 | (-0.047, 0.328) | 0.0730 | (-0.123, 0.269) |
| poverty rate | -0.2701 | (-0.809, 0.269) | -0.7307 | (-1.622, 0.161) | 0.0286 | (-0.645, 0.702) | 0.0613 | (-0.607, 0.729) |
| log median house value | -0.0414 | (-0.151, 0.068) | 0.2398 | (-0.03, 0.509) | 0.2781 | (0.013, 0.543) | 0.3458 | (0.075, 0.616) |
| percent black | 0.2393 | (-0.086, 0.565) | 1.3918 | (0.539, 2.245) | 0.2837 | (-0.099, 0.666) | 0.5388 | (0.144, 0.934) |
| log median household income | 0.0284 | (-0.224, 0.281) | -0.5573 | (-1.451, 0.337) | -0.6099 | (-1.062, -0.158) | -0.7041 | (-1.178, -0.231) |
| percent owner occupied housing | 0.3923 | (0.013, 0.772) | 1.0197 | (-0.412, 2.451) | 0.7434 | (0.382, 1.104) | 0.6334 | (0.28, 0.987) |
| percent hispanic | -0.5473 | (-1.065, -0.029) | -1.0773 | (-1.787, -0.368) | 0.6337 | (0.038, 1.23) | 0.6459 | (0.039, 1.253) |
| percent less than HS education | 0.6412 | (0.307, 0.976) | 0.3386 | (-0.505, 1.182) | 0.0844 | (-0.433, 0.602) | 0.1517 | (-0.349, 0.653) |
| 2nd quintile pop. density | 0.0648 | (0.004, 0.125) | -0.0905 | (-0.166, -0.015) | 0.0651 | (-0.035, 0.165) | -0.0083 | (-0.103, 0.087) |
| 3rd quintile pop. density | 0.1330 | (0.079, 0.187) | 0.0199 | (-0.045, 0.085) | 0.1226 | (0.015, 0.23) | 0.0422 | (-0.056, 0.14) |
| 4th quintile pop. density | 0.1206 | (0.075, 0.166) | 0.0310 | (-0.037, 0.099) | 0.1916 | (0.032, 0.351) | 0.0882 | (-0.056, 0.232) |
| 5th quintile pop. density | 0.1499 | (0.099, 0.201) | 0.0685 | (-0.07, 0.207) | 0.2359 | (0.067, 0.405) | 0.1170 | (-0.035, 0.269) |
| $\log(y^*_{i,t}/m_i)$ | 0.6822 | (0.646, 0.719) | 0.6433 | (0.61, 0.676) | 0.5010 | (0.461, 0.541) | 0.5015 | (0.464, 0.539) |

**Table 6. Coefficients and 95% confidence intervals based on robust standard errors for the excess zero model.**

| | Hurdle-NB | | GEE-ZIOP | |
|---|---|---|---|---|
| | λ | CI | λ | CI |
| Intercept | -9.5816 | (-10.5353, -8.628) | -5.4241 | (-6.46, -4.3882) |
| holiday tomorrow | -0.5704 | (-1.1793, 0.0386) | -0.3106 | (-1.2357, 0.6145) |
| holiday today | -0.2420 | (-0.9197, 0.4357) | -0.5812 | (-1.1974, 0.0349) |
| holiday yesterday | -0.3922 | (-1.0473, 0.2629) | 0.3125 | (-1.1039, 1.7289) |
| weekend | 0.3045 | (0.0086, 0.6003) | 0.4318 | (-0.0858, 0.9495) |
| poverty rate | 6.5020 | (5.1133, 7.8907) | 7.3636 | (5.8934, 8.8338) |
| percent owner occupied housing | 8.8344 | (7.6426, 10.0262) | 3.8253 | (2.9493, 4.7013) |
| $\mathbb{I}\left(y_{i,t-1}=0\right)$ | 2.4586 | (2.2854, 2.6318) | 1.5558 | (1.4174, 1.6942) |

GEE-ZIOP: $\tau_Z$ = 197439.4

Hurdle-NB: negative binomial Hurdle model, without spatial correlation

GEE-ZIOP: zero-inflated over-dispersed Poisson marginal model, with spatial correlation

increased probability of reporting zero cases. This confounding may be due to logistical challenges in reporting cases at a daily frequency and less community transmission due to sparse population density in rural areas. In addition, high poverty rate was significantly associated with an increased probability of reporting zero cases in both models. By testing the effects of socioeconomics factors on case reporting, we found that zero reported cases could be confounded by disease surveillance resources of local municipalities. Rural and poor counties may lack necessary resources to reliably report cases leading to excess zeros which can ultimately hinder the modeling COVID-19 incidence rates if not addressed.

## Discussion

One of the goals of this analysis was to understand the relationship between holiday timing and surges in COVID-19 cases while accounting for zero-inflation, temporal effects, and spatial correlations in case counts from neighboring areas. We implemented an ES algorithm for the estimation of our zero-inflated Poisson model as a mixture of marginal models. We used a semivariogram for the spatial working correlation and update it in the S-step, after each Newton Raphson iteration. We take advantage of fast and robust semivariogram estimation as a sub-routine in the greater ES algorithm. Considering the zero-inflated nature of the data, carried forward imputation and time-lagged covariates were used to account for the temporal pattern of case reporting. After the estimations have converged, we used the robust covariance estimator, which is consistent under misspecified spatial correlation, to compute standard errors.

We analyzed data from March 20, 2020 to January 23, 2021, before widespread vaccine distribution, COVID-19 variants and at-home testing. Our analysis suggests a statistically significant surge in reported cases, an IRR of 1.3 to 1.41, 6-8 days after a holiday, with the surge of cases gradually tapering off afterward. This is in line with the anticipated timing of 4–5 days to symptom onset and up to 3 days for PCR testing results. Although the models estimated the surge in reported cases to be 6-8 days after a holiday, transmission may have occurred days prior to the holiday. It's important to note that a calendar holiday date is not the exact date of COVID-19 transmission. For example, traveling may commence the Friday before a Monday holiday or many people may also extend their holidays so people travel and congregate days after a holiday, e.g., Thanksgiving and Friday holidays. The reality is that there is a window of

days surrounding a holiday which the transmission likely occurred. In contrast, we observed negative effects 2–3 days after a holiday which could be explained by testing lag associated with holidays. If one contracts COVID-19 when traveling near the calendar holiday date, it takes on average 4–5 days for symptoms to appear. We believe that the negative effect timing could be explained by the asymptomatic incubation period right after contracting COVID-19 when people have yet to be tested.

All but the hurdle model, estimates in Fig 2 confirm a drop in the case counts on a holiday, with many of the backlogged cases being reported the next day, evident by a large positive effect one day after a holiday. In our GEE-ZIOP model, the IRR is estimated to be 0.86 when the current is a holiday and 1.31 one day after a holiday. This trend is aligned with case reporting guidelines. Note that all four models (GLM, Hurdle-NB, GEE-OP and GEE-ZIOP) share similar conclusions regarding holiday effect, indicating the surge in reported cases occurs 6-8 after a holiday, suggesting that these results are robust under different assumptions.

Comparing different models, the GEE-ZIOP model was associated with an increase in the coefficient estimates, but at the expense of efficiency. When using GEE-ZIOP, weighting each data point by the zero-inflation membership probability decreases the effective sample size. However, even when considering this trade-off, there are often still meaningful results; our GEE-ZIOP estimate for the coefficients of HL6, HL7, and HL8 are significantly greater than zero. Furthermore, the GEE-ZIOP model detected an IRR significantly greater than 1, 6 days after a holiday, while GEE-OP did not. Several holidays (Memorial, Labor, Columbus) fall on a Monday, thus 6 days after a Monday is a Sunday and Sundays are likely dates where zero cases are reported. GEE-ZIOP allows us to down weight these excess zeros, revealing that surges in reported cases can occur as early as 6 days after a holiday.

In summary, we expect there to be a surge in county level reported cases one week after a holiday in the state of Pennsylvania. While our results reaffirm the common intuition regarding the early COVID-19 pandemic, our regression model elucidates the temporal trends of post-holiday case counts over a two-week period. In conclusion, we present results that illustrate the timing of a post-holiday surge while considering federal holidays and election day. Understanding the post-holiday surges can inform public policy and can be used to improve public health programs. We believe that our modeling approach and conclusions based on the rich data gathered through rigorous surveillance during the beginning of the pandemic, can guide future disease modeling analyses and are applicable to a wide range of epidemiological settings.

## Author Contributions

**Conceptualization:** Benny Ren, Wei-Ting Hwang.

**Data curation:** Benny Ren, Wei-Ting Hwang.

**Formal analysis:** Benny Ren, Wei-Ting Hwang.

**Funding acquisition:** Wei-Ting Hwang.

**Investigation:** Benny Ren, Wei-Ting Hwang.

**Methodology:** Benny Ren, Wei-Ting Hwang.

**Project administration:** Benny Ren, Wei-Ting Hwang.

**Resources:** Benny Ren, Wei-Ting Hwang.

**Software:** Benny Ren, Wei-Ting Hwang.

**Supervision:** Benny Ren, Wei-Ting Hwang.

**Validation:** Benny Ren, Wei-Ting Hwang.

**Visualization:** Benny Ren, Wei-Ting Hwang.

**Writing – original draft:** Benny Ren, Wei-Ting Hwang.

**Writing – review & editing:** Benny Ren, Wei-Ting Hwang.

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
