## [Decision Letter · Decision Letter 0]

20 Sep 2022

PONE-D-22-12140Modeling Post-Holiday Surge in COVID-19 Cases in Pennsylvania CountiesPLOS ONE

Dear Dr. Ren,

Thank you for submitting your manuscript to PLOS ONE. After careful consideration, we feel that it has merit but does not fully meet PLOS ONE’s publication criteria as it currently stands. Therefore, we invite you to submit a revised version of the manuscript that addresses the points raised during the review process.

We look forward to receiving your revised manuscript.

Kind regards,

Chong Wang, Ph.D.

Academic Editor

PLOS ONE

Journal Requirements:

“WH is supported by National Institute of Environmental Health Sciences grant: P30-ES013508. The funders had no role in study design, data collection and analysis, decision to publish, or preparation of the manuscript.”

Additional Editor Comments (if provided):

Both reviewers recommended "major revisions". Please address their comments accordingly.

Reviewers' comments:

Reviewer's Responses to Questions

**Comments to the Author**

1. Is the manuscript technically sound, and do the data support the conclusions?

Reviewer #1: Yes

Reviewer #2: Partly

2. Has the statistical analysis been performed appropriately and rigorously? 

Reviewer #1: Yes

Reviewer #2: No

3. Have the authors made all data underlying the findings in their manuscript fully available?

Reviewer #1: No

Reviewer #2: Yes

4. Is the manuscript presented in an intelligible fashion and written in standard English?

Reviewer #1: Yes

Reviewer #2: Yes

5. Review Comments to the Author

Reviewer #1: Major Point:

The first thing to point out is that the problem the authors address is not new. Researchers have been doing this sort of spatial analysis for quite some time. See the books by Lawson (2018), Blangiardo and Cameletti (2015), Moraga (2019), and Banerjee et al. (2015).

Lawson, A. B. (2018). Bayesian disease mapping: hierarchical modeling in spatial epidemiology. Chapman and Hall/CRC.

Blangiardo, M., & Cameletti, M. (2015). Spatial and spatio-temporal Bayesian models with R-INLA. John Wiley & Sons.

Moraga, P. (2019). Geospatial health data: Modeling and visualization with R-INLA and shiny. CRC Press.

Banerjee, S., Carlin, B. P., & Gelfand, A. E. (2015). Hierarchical modeling and analysis for spatial data, Second edition. Chapman and Hall/CRC.

The solution that most authors apply is the conditional autoregressive (CAR) model, which assumes a covariance structure that depends on the network of neighbors (which can be defined in a number of ways). Data are almost always analyzed in the Bayesian context, using either Markov chain Monte Carlo (MCMC) or the integrated nested Laplace approximation (INLA).

The authors propose a model for the covariance matrix that depends on the distances (through the semivariogram). This approach is very close to the concept of kriging, a technique mostly used to interpolate data (such as air pollution measurements) taken at only a few locations across a grid. Predictor variables can be used with kriging, similar to how the authors have used them in the paper under review. The only mention of "kriging" in the paper comes from the article title in reference [36] by Dowd. Lawson (2018), in Section 5.4.2, mentions how a fully specified covariance matrix, like the authors, can be used for areal data, where the measurements are the county-level outcomes at the centroid of the region. Lawson says this approach is "akin to universal kriging."

The authors' approach uses generalized estimating equations, a classical approach. This is certainly different from the usual approach in the four references described above. Is the authors' approach better than the Bayesian approaches described in the four references above? The answer might be no. But, if the answer is yes, then the paper may be more impactful than the authors realize.

Other points:

1. Be more thorough on lag between contracting COVID and testing positive. There is some science here, especially regarding latency. Local conditions affect time to get tested and reporting lag.

2. Be careful with structural zeros. The probabilistic model you suggest may not be reasonable. In my neck of the woods, some health departments never on Saturdays, Sundays, and holidays. Some rarely report on Mondays, presumably because the large backlog takes more than one day. If your model proposes a probability of reporting a zero on a given day, then you may be predicting low probabilities for some counties when the probability should be zero.

3. In the equation at the bottom of p. 3, I expect to see Y_{i,t-1} on the right side. This is, after all, the conditional distribution of Y_{i,t} given Y_{i,t-1}. It's not until the next paragraph (on the next page) that we see that mu_i depends on Y_{i,t-1} .

4. What happens to equation (1) when y_{i,t-1} = 0. The log would be undefined. The next equation suggests that the whole expression should be 0.

5. In Equation (2), the probability on the left conditions on Z_{i,t-1} (cap Z) while the right side involves z_{i,t-1} (lower case z).

6. The section on the "Theoretical Semivariogram Model" begins by talking about the residuals. But residuals to what model? There seems to be a Catch-22 here. Residuals from the model are needed to define the model. Is there a way out of this circle?

7. p. 3, second last line. This should be "While standard GEEs treat longitudinal data ..."

8. Figure 1. I don't understand what this plot is showing. The caption describes what the dashed line means, but what are the boxplots on? And shouldn't the dashed lines go through the centers of the boxes? I'm probably missing an important idea, but the caption doesn't help.

9. p. 5, just above the equation in the middle of the page. The colons (:) on the second and first lines above the equation for phi are not needed. Also, in the equation for phi, go ahead and write the indices for i: \\sum_{i=1}^{67}, or if you want to keep it more general, introduce a variable name for the number of regions and use that.

10. p. 6, line 110. I believe Z_{t-1} should be Z_{i,t-1}.

11. There are exactly S choose 2 unique distances and exactly S choose 2 times T computations of the differences of residuals. Don't interject big O notation if it isn't necessary.

12. Be consistent on the citations. Mostly, the authors use the reference number in brackets, e.g. [25], but sometimes they use the name year. At the top of the page, they refer to "Cressie and Hawkins, 1980" which I think should be reference [34]. Later we see Hall and Zhang, 2004.

13. If I'm interpreting Figure 2 right, there are some negative correlations when days since a holiday is small and positive correlations about a week after the holiday. I suspect the former is due to a reporting lag, and the latter is due to the holiday effect. Is this correct? A little more explanation would help here.

14. In the last paragraph on p. 13, the authors say there is an IRR of 1.3 to 1.4. It seems to me that this is the number for the state as a whole. What can you say about individual counties? Is this surge higher in some counties than others? Showing maps would be helpful here.

15. In line 361, the authors say "All but the hurdle model, estimates in Figure 2 confirm a drop in the case counts on a holiday, with all the backlogged cases being reported the next day." I think it is optimistic to say that ALL backlogged cases are reported the next day. Around here, this is certainly not the case.

16. The section on Supporting Information seems to be blank.

17. Be consistent in capitalizing journal names. Either capitalize the first letters of important words always or never. For example, we have "New England journal of medicine" but "International Journal of Infectious Diseases." On p. 16, we see "American journal of public health" and "Journal of the American Statistical Association." Also, "Jama" looks odd.

Reviewer #2: The introduction of the proposed method is clear. However, there are some points that are not reasonable or need to be further explained. Since these data were obtained before widespread vaccine distribution, now the vaccine is widely distributed in many countries, the conclusions of this research problem do not have high guiding value for the actual situation. Please see the referee report for detailed comments.

6. PLOS authors have the option to publish the peer review history of their article (what does this mean?). If published, this will include your full peer review and any attached files.

Reviewer #1: No

Reviewer #2: No

---

## [Author Response · Author response to Decision Letter 0]

1 Nov 2022

Please see the "Response to Reviewers" document.

---

## [Decision Letter · Decision Letter 1]

6 Dec 2022

Modeling Post-Holiday Surge in COVID-19 Cases in Pennsylvania Counties

PONE-D-22-12140R1

Dear Dr. Ren,

We’re pleased to inform you that your manuscript has been judged scientifically suitable for publication and will be formally accepted for publication once it meets all outstanding technical requirements.

Kind regards,

Chong Wang, Ph.D.

Academic Editor

PLOS ONE

Additional Editor Comments (optional):

Reviewers' comments:

Reviewer's Responses to Questions

**Comments to the Author**

1. If the authors have adequately addressed your comments raised in a previous round of review and you feel that this manuscript is now acceptable for publication, you may indicate that here to bypass the “Comments to the Author” section, enter your conflict of interest statement in the “Confidential to Editor” section, and submit your "Accept" recommendation.

Reviewer #2: All comments have been addressed

2. Is the manuscript technically sound, and do the data support the conclusions?

Reviewer #2: Yes

3. Has the statistical analysis been performed appropriately and rigorously? 

Reviewer #2: Yes

4. Have the authors made all data underlying the findings in their manuscript fully available?

Reviewer #2: Yes

5. Is the manuscript presented in an intelligible fashion and written in standard English?

Reviewer #2: Yes

6. Review Comments to the Author

Reviewer #2: I am satisfied with the revision and responses to my comments. Please make sure there are no grammatical or spelling errors in your paper.

7. PLOS authors have the option to publish the peer review history of their article (what does this mean?). If published, this will include your full peer review and any attached files.

Reviewer #2: No

---

## [Editor Report · Acceptance letter]

9 Dec 2022

PONE-D-22-12140R1 

Modeling post-holiday surge in COVID-19 cases in Pennsylvania counties 

Dear Dr. Ren:

I'm pleased to inform you that your manuscript has been deemed suitable for publication in PLOS ONE. Congratulations! Your manuscript is now with our production department. 

Kind regards, 

on behalf of

Professor Chong Wang 

Academic Editor

PLOS ONE